# From Deworming to Cancer Therapy: Benzimidazoles in Hematological Malignancies

**DOI:** 10.3390/cancers16203454

**Published:** 2024-10-12

**Authors:** Upendarrao Golla, Satyam Patel, Nyah Shah, Stella Talamo, Riya Bhalodia, David Claxton, Sinisa Dovat, Arati Sharma

**Affiliations:** 1Division of Hematology and Oncology, Department of Medicine, Pennsylvania State University College of Medicine, Hershey, PA 17033, USA; ugolla@pennstatehealth.psu.edu (U.G.); dclaxton@pennstatehealth.psu.edu (D.C.); 2Penn State Cancer Institute, Pennsylvania State University College of Medicine, Hershey, PA 17033, USA; 3Department of Pharmacology, Pennsylvania State University College of Medicine, Hershey, PA 17033, USA; spatel22@pennstatehealth.psu.edu; 4Department of Health Sciences, McMaster University, Hamilton, ON L8S 4L8, Canada; nyah.shah@mail.utoronto.ca; 5Department of Medicine, Liberty University College of Osteopathic Medicine, Lynchburg, VA 24502, USA; stalamo@liberty.edu; 6Division of Hematology and Oncology, Department of Pediatrics, Pennsylvania State University College of Medicine, Hershey, PA 17033, USA; riyabhalodia2@gmail.com (R.B.); sdovat@pennstatehealth.psu.edu (S.D.)

**Keywords:** cancer, drug repurpose, anthelmintic, benzimidazole derivatives, signaling pathways, leukemia, lymphoma, tubulin polymerization, apoptosis, differentiation therapy

## Abstract

**Simple Summary:**

Drug repurposing offers a valuable approach for utilizing FDA-approved drugs to address new diseases, including cancer, while significantly reducing the time and cost associated with drug development. In this review, we highlight the anticancer potential of anthelmintic benzimidazole derivatives, such as mebendazole (MBZ), fenbendazole (FBZ), flubendazole (FLBZ), and albendazole (ABZ), with a particular focus on their efficacy against blood cancers. These compounds operate through multiple mechanisms, including disrupting microtubule formation, regulating the cell cycle, inducing cellular differentiation, and modulating key signaling pathways involved in cancer progression. The ability of benzimidazoles to exert these diverse effects on malignant cells suggests their promise as therapeutic agents for hematological cancers. Furthermore, investigating their use in combination with standard-of-care treatments may lead to the development of novel, more effective therapeutic strategies, offering new avenues of hope for patients with blood malignancies.

**Abstract:**

Drug repurposing is a strategy to discover new therapeutic uses for existing drugs, which have well-established toxicity profiles and are often more affordable. This approach has gained significant attention in recent years due to the high costs and low success rates associated with traditional drug development. Drug repositioning offers a more time- and cost-effective path for identifying new treatments. Several FDA-approved non-chemotherapy drugs have been investigated for their anticancer potential. Among these, anthelmintic benzimidazoles (such as albendazole, mebendazole, and flubendazole) have garnered interest due to their effects on microtubules and oncogenic signaling pathways. Blood cancers, which frequently develop resistance and have high mortality rates, present a critical need for effective therapies. This review highlights the recent advances in repurposing benzimidazoles for blood malignancies. These compounds induce cell cycle arrest, differentiation, tubulin depolymerization, loss of heterozygosity, proteasomal degradation, and inhibit oncogenic signaling to exert their anticancer effects. We also discuss current limitations and strategies to overcome them, emphasizing the potential of combining benzimidazoles with standard therapies for improved treatment of hematological cancers.

## 1. Introduction

Despite decades of efforts aimed at finding effective treatments, cancer remains the leading cause of death worldwide. According to the World Health Organization (WHO), there were an estimated 20 million new cancer cases and 9.7 million deaths in the year 2022. The projected rise in cancer cases by 2050 is approximately 35 million based on the projected population. These estimates suggest that almost one in five humans will develop cancer in a lifetime, whereas about one in nine men and one in 12 women die from it [1]. Recent statistics from 2024 estimate that there will be approximately 2 million new cancer cases diagnosed and 611,720 cancer-related deaths in the United States alone [2]. Hematological malignancies account for the fourth most common type of cancer and the leading cause of deaths. Hematological cancers include leukemias (ALL-acute lymphoblastic leukemia, CLL-chronic lymphoid leukemia, AML-acute myeloid leukemia, CML-chronic myeloid leukemia), lymphomas (Hodgkin’s and non-Hodgkin’s type), and multiple myeloma (MM) [3]. Half of these blood malignancies are diagnosed in patients 65 years and older, and 70% of deaths due to cancer occur in this population. It is expected that there will be a 68% increase in the incidence of cancer by 2030, due to the aging of the U.S. population and a lack of precise treatment options [4]. Although new anticancer agents are discovered each year, the overall rise in cancer incidence, drug resistance, and toxicity impose great hurdles for the treatment of cancer. Therefore, there is a need for innovative approaches such as repurposing existing drugs as new effective agents targeting cancers alone or in combination with standard therapies [5,6].

Drug repurposing, also called repositioning, is an approach for discovering new uses for approved drugs or drugs that failed clinical trials due to a lack of efficacy. Such repurposing has several advantages over developing an altogether new drug, the two most important being low risk of failure due to pre-established safety and less time in development [7]. Another advantage is less investment in repurposing the therapeutic indication of an already approved drug. It is estimated that approval of drugs through the repurposing route takes an average of 3–12 years and costs around $40–80 million, which is significantly lower than conventional drug discovery [8]. Among the many different chemical classes of drugs, benzimidazole derivatives that are traditionally recognized as anthelmintic agents have recently gained the attention of oncology researchers due to their potential anticancer activities [9,10]. Due to their inhibitory effects on tubulin polymerization, which in turn leads to cell death due to mitotic arrest, benzimidazole derivatives can serve as attractive agents to repurpose for treatment of cancers. Several benzimidazole derivatives have undergone clinical trials in different cancer types, such as liver, lung, breast cancer, etc. [9,11]. Alongside their anticancer activity, these anthelmintic agents also influence immune responses by modulating a specific subset of immune cells and related pathways [12]. Although benzimidazole-based derivatives were recently approved as anticancer agents [13], few agents such as Bendamustine (NCT04510636), Pevonedistat (NCT00911066), and Pracinostat (NCT03848754) are under clinical investigation for the treatment of leukemia and lymphoma [14,15,16]. The use of these benzimidazole derivatives for hematological cancers is still under development, and further evaluation of mechanisms underlying antileukemic effects is needed.

Though several recent reviews have summarized the anticancer potential of these benzimidazole derivatives [9,11,17], there is a need for a focused literature review detailing their antileukemic effects. This review highlights the emerging approach of repurposing benzimidazole derivatives for the treatment of various blood malignancies alone or in combination with other standard of care (SOC) therapies. To this end, we summarize multiple studies that have repurposed anthelmintic benzimidazoles for the treatment of blood cancers in preclinical models. We also discuss various molecular mechanisms that contribute to the antileukemic activity exerted by different FDA-approved anthelmintic benzimidazoles such as albendazole, mebendazole, flubendazole, and fenbendazole.

## 2. Benzimidazoles as Anthelmintic Agents

Benzimidazoles are structurally classified as heterocyclic aromatic compounds in which a six-membered benzene ring is fused with a five-membered imidazole. Since benzimidazoles gained popularity in the early 1960s as anthelmintic agents, they have been widely used in both veterinary and human medicine to treat gastrointestinal infections caused by parasitic worms [18]. Several benzimidazoles, including mebendazole, flubendazole, nocodazole, albendazole, and fenbendazole, have been very successful as anthelmintic agents (Figure 1). Benzimidazoles mechanism of action as anthelmintic agents involves selective binding to β-tubulin, a microtubule subunit crucial for parasitic functions such as cell division and organelle transport. By inhibiting tubulin polymerization and disrupting microtubule formation, benzimidazoles lead to parasite destruction. Additionally, they interfere with glucose transport and metabolism in parasites, disrupting energy metabolism, which also aids in parasite death [19].

Since its first synthesis in 1872, benzimidazoles have attracted growing interest due to extensive research demonstrating the interaction of heterocyclic moiety with a wide range of unrelated biological molecular targets. Figure 2 illustrates the wide spectrum of biological activities associated with benzimidazole-based derivatives, underscoring their versatility and therapeutic promise across various medical domains. In addition to anthelmintic activity, benzimidazole-based derivatives are established as antimicrobials, antifungals, antivirals, anticonvulsants, antihypertensives, analgesics, antihistaminics, anti-inflammatory agents, anticoagulants, antiulcer agents, and anticancer agents [13,20]. Current research endeavors focus on elucidating the potential of chemical structure, synthesis, biological applications, and drug repurposing of benzimidazole derivatives as anticancer agents through their effects on microtubules and oncogenic pathways.

## 3. Repurposing of Anthelmintic Benzimidazoles as Anticancer Agents

Benzimidazole-based derivatives exert anticancer activity through regulation of several important oncogenic signaling pathways and cellular processes such as cell cycle, autophagy, differentiation, metastasis, angiogenesis, etc. (Figure 3) [13,17,21]. Although a countless number of benzimidazole derivatives are in the preclinical stages, to date, only bendamustine has received FDA approval as an anticancer agent to treat CLL, MM, and non-Hodgkin’s lymphoma [22]. In a study conducted by Knauf et al., involving 319 patients aged 75 or younger, bendamustine administered intravenously was compared with orally administered chlorambucil for efficiency and safety in untreated patients with advanced-stage CLL. Results indicated that bendamustine exhibited a higher complete or partial response rate of 68%, compared to 31% with chlorambucil. Additionally, bendamustine showed a superior median progression-free survival rate of 21.6 months, compared to 8.3 months with chlorambucil, along with a longer duration of remission at 21.8 months as opposed to 8.0 months with chlorambucil. These results underscore bendamustine’s efficacy and favorable toxicity profile, establishing it as the primary treatment drug for patients with advanced CLL [23].

Furthermore, Leoni et al. examined and characterized the mechanisms of action of bendamustine, a previously profiled compound in the National Cancer Institute’s (NCI) In vitro Cell Line Screening Project (IVCLSP), which determines a compound’s activity pattern using sixty different human tumor cell lines. Data from gene microarray profiles, quantitative PCR (qPCR), immunoblot, cell cycle analyses, and microscopy was further utilized to determine and compare bendamustine’s mechanism of action to other DNA alkylating agents, such as chlorambucil [24]. The results indicate that bendamustine displays various other mechanisms of actions unrelated to other alkylating agents, such as activation of DNA damage stress response and apoptosis, specifically in non-Hodgkin’s lymphoma cells, inhibition of mitotic checkpoints, and induction of mitotic catastrophe. In terms of DNA damage repair and apoptosis, p53 is a vital tumor suppressor protein that regulates apoptosis in the event of DNA damage or other cellular catastrophes, with a key event in the process being phosphorylation at Ser15. Leoni et al. found that bendamustine resulted in an 8-fold up-regulation of Ser15-phosphorylated p53 and induction of p53-dependent genes in cells, thus triggering the p53-dependent stress pathway that results in a strong activation of intrinsic apoptosis. Strong activation of apoptosis in conjunction with the inhibition of mitotic checkpoints, which results in the induction of mitotic catastrophe in cells entering mitosis with extensive DNA damage, could explain why bendamustine is effective in drug-resistant cells in vitro and lymphoma patients with chemotherapy-refractory disease [24].

Other promising benzimidazole-based anticancer drugs that are advanced to clinical trials include Selumetinib (NCT02768766), Binimetinib (NCT04965818 and NCT03170206), Abemaciclib (NCT04003896 and NCT0404-0205), Dovitinib (NCT01635907), Veliparib (NCT02723864 and NCT01434316), Pracinostat (NCT03848754), Nazartinib (NCT02335944 and NCT02108964), and Galeterone (NCT04098081) [13]. Both Selumetinib and Galeterone are on the edge of FDA approval due to ongoing phase II/III clinical trials. Selumetinib is a selective inhibitor of mitogen-activated protein kinase (MAPK)/extracellular signal-related kinase (ERK) kinases 1 and 2 (MEK1 and MEK2); it displayed its potency as an antiproliferative agent when treating different human cancer cell lines, such as NSCLC, melanoma, pancreatic, and colorectal cell lines. Further analysis revealed that cell lines containing mutant BRAF and RAS were susceptible to selumetinib. Selumetinib’s mechanism of action starts with selectively and effectively inhibiting ERK1 and ERK2, substrates of MEK1 and MEK2 in the MAP kinase pathway. Galeterone, initially designed to inhibit androgen biosynthesis, was further shown to regulate two more targets in the androgen/androgen receptor pathway and inhibit the eukaryotic initiation factor 4E (eIF4E) protein translational machinery. Galeterone successfully passed clinical trial phases I and II in patients with prostate cancer but could not pass phase III trials in men with castration-resistant prostate cancer (CRPC), harboring AR splice variants (e.g., AR-V7). Additionally, a recent review by Song et al. highlighted various clinical findings that explore the anticancer potential of traditional anthelmintic benzimidazoles through drug repurposing strategies [9]. Therefore, the benzimidazole-based derivatives gained attention in recent years for repurposing to treat different human malignancies.

## 4. Repurposing of Anthelmintic Benzimidazoles for the Treatment of Hematological Cancers

Hematological cancers begin in the cells of the immune system or blood-forming tissue, such as the bone marrow, and disrupt the normal function of the hematopoietic system. Genomic analysis and molecular characterization play a vital role in the diagnosis and clinical management of almost every form of hematological malignancy. In general, these blood cancers are grouped as leukemia, lymphoma, and multiple myeloma. Based on the genetic analysis, these malignancies are highly heterogeneous and subclassified as acute leukemias, myelodysplastic syndromes (MDS), myeloproliferative neoplasms (MPNs), non-Hodgkin lymphomas, and classical Hodgkin [25,26] lymphomas. Although over 50 agents have been approved by the United States Food and Drug Administration for the treatment of hematological cancers, there remains a significant need for more targeted, safer, and effective therapies to address emerging genetic mutations, cancer stem cells, and drug resistance pathways [27] In this regard, drug repurposing presents an appealing and effective strategy to discover additional treatments that can work independently or in combination with standard therapies. In hematological cancers, the repurposing of anthelmintic benzimidazoles has emerged as a promising avenue for novel and less toxic therapeutic interventions. This comprehensive overview explores the potential of Mebendazole (MBZ), Albendazole (ABZ), Flubendazole (FLBZ), and Fenbendazole (FBZ) across diverse hematological malignancies, including AML, ALL, MM, CML, and lymphoma. From potent effects in AML to differentiation therapy in non-APL AML, MBZ and ABZ demonstrate efficacy with minimal impact on normal cells. In ALL, MBZ’s ability to suppress proliferation and induce cell cycle arrest in chemoresistant scenarios is highlighted. In Multiple Myeloma (MM), these benzimidazoles exhibit growth-inhibitory effects and interfere with critical oncogenic pathways. CML benefits from MBZ’s capacity to overcome drug resistance and induce apoptosis. Moreover, in lymphomas, FLBZ and FBZ showcase anti-lymphoma activity, emphasizing the potential of these compounds across various hematological cancers. This condensed overview and Table 1 underscore the multifaceted potential of repurposed anthelmintic benzimidazoles, promising a new era in hematologic cancer therapeutics with improved efficacy and reduced toxicity.

### 4.1. Acute Myeloid Leukemia (AML)

AML is a malignancy characterized by abnormally differentiated hematopoietic cells that infiltrate the bone marrow, blood, and other tissues. Cytarabine-based chemotherapy has been the standard treatment for AML for decades; however, significant adverse effects, such as myelosuppression and infection, frequently result from its cytotoxicity to both healthy and malignant hematopoietic cells. As such, a major aim of a multitude of studies is to identify novel compounds that can effectively treat AML [43]. Recent studies have revealed the potential of benzimidazole-containing compounds in the treatment of AML.

#### 4.1.1. Mebendazole

In numerous studies, the anthelmintic drug MBZ, a benzimidazole-containing derivative, demonstrated potent effects as a potential new low-toxicity therapeutic for human AML. At pharmacologically feasible doses, MBZ was reported to impair the development of AML cell lines and bone marrow mononuclear cells from AML patients [36]. MBZ also induces mitotic arrest and mitotic catastrophe in AML cells while repressing the progression of leukemic cells in vivo and prolonging survival in AML xenograft mouse models [36,37]. Furthermore, He et al. found MBZ to have little inhibitory effect on normal peripheral blood mononuclear cells or human umbilical vein endothelial cells, suggesting its sufficiency in inhibiting colony formation of AML cells while eliciting low toxicity to normal cord blood-derived cells [36].

The transcription factor c-MYB has been found to play a vital role in the progression of AML through various oncogenes, such as mixed lineage leukemia (MLL)-fusion genes [44]. Walf-Vorderwulbecke et al. identified MBZ as capable of interfering with c-MYB-regulated transcriptional pathways in human AML cells by inhibiting protein folding through blockade of the heat shock protein 70 (HSP70) chaperone system [38]. Results showed that colony formation of THP1 AML cells was reduced by more than 80% while having no effect on normal CD34+ cord blood cells. Bioluminescence imaging further showed in vivo MBZ activity to inhibit leukemia progression in luciferase-expressing THP1 AML murine xenografts, extending the survival of treated mice compared to the control.

It is well established that GLI transcription factors help to sustain leukemia by initiating chemotherapy-resistant cells leading to therapy failure and tumor relapse [38]. As such, Freisleben et al. showed that MBZ mediates strong anti-leukemic effects by promoting the proteasomal degradation of GLI transcription factors through inhibition of HSP70/90 chaperone activity and sensitizing AML cells to chemotherapy [39,45]. These findings suggest that MBZ can be repurposed for evaluation in AML treatment.

#### 4.1.2. Albendazole

Compared to intensive chemotherapy, a less toxic but still highly effective treatment for acute promyelocytic leukemia (APL) AML is differentiation therapy, which results in the degradation of APL cells to neutrophils with subsequent leukemia cell death [22,23,24,46]. However, no effective differentiation chemotherapy for non-APL leukemia exists. ABZ is a commonly used anthelmintic drug shown to be an effective anti-leukemia agent. As such, Noura et al. performed a chemical screening and identified ABZ as the most promising compound to differentiate non-APL AML cells to monocytes through stimulation of the Krüppel-like factor 4 (KLF4)-dihydropyrimidinase-like 2A (DPYSL2A) differentiation axis [28]. Based on in vitro and in vivo tests, ABZ has been determined as an emerging candidate drug for differentiation chemotherapy for patients with non-APL AML [28]. Similarly, after having performed an FDA drug repurposing screen using 760 drugs, Matchett et al. also found ABZ to have particular anti-leukemic efficacy in primary cells and cell lines representing a range of molecular subtypes of AML, whilst having negligible effect in normal murine bone marrow cells, suggesting low toxicity [18].

#### 4.1.3. Flubendazole and Fenbendazole

Other anthelmintic compounds tested in leukemia cells were FLBZ and FBZ. FLBZ has been found to induce cell death in leukemia and myeloma cell lines and primary patient samples at nanomolar concentrations, showing no signs of toxicity in the process [34]. In terms of its mechanism, FLBZ alters microtubule structure and inhibits tubulin polymerization by interacting with a different site on tubulin than that of vinblastine, a chemotherapy medication. In addition, cells that were resistant to vinblastine were found to be fully sensitive to FLBZ, indicating that FLBZ can overcome some forms of vinblastine resistance. The combination of FLBZ and vinblastine was synergetic in nature, showing a reduction in the viability of OCI-AML2 cells. In addition, combinations of FLBZ with vinblastine or vincristine in a leukemia xenograft model delayed tumor growth more than when the drugs were acting alone [34].

Differentiation therapy is a new treatment option for some cancer types, as it rapidly and irreversibly clears tumor bulk following terminal maturation of blast cells. KalantarMotamedi et al. showed that FBZ can induce cell death of HL60 leukemia cells within a short period of 3 days via induction of differentiation to granulocytes in low concentrations of 0.1 μM [31]. This was confirmed by cellular imaging of HL60 cells with FBZ treatment, as well as a nitroblue tetrazolium reduction assay, which indicated a transformation of leukemia cells to granulocytes upon neutrophil differentiation. Furthermore, following 72 h of treatment, FBZ exhibited a 14.5-fold higher selectivity in targeting and killing HL60 cells compared to human bone marrow stem cells [31].

### 4.2. Acute Lymphoblastic Leukemia (ALL)

ALL is a hematological malignancy characterized by uncontrolled proliferation of abnormal, immature lymphocytes that permeate the bone marrow and blood [47]. T-cell acute lymphoblastic leukemia (T-ALL) currently represents 15% of all ALL cases in children and 25% in adults, drawing a lot of attention to researching new treatment interventions [48]. Currently, conventional chemotherapy therapeutics such as doxorubicin, paclitaxel, vincristine, and dexamethasone are used in the treatment of ALL; however, these drugs often result in higher relapse rates and drug resistance [49].

#### Mebendazole

Wang et al. found MBZ to suppress the proliferation and reduce the viability of T-ALL cell lines in a dose-dependent manner [40]. DAPI staining and flow cytometry analysis showed MBZ to induce dose-dependent cell cycle arrest at G2/M phase at 24 h. Caspase 3/7 activity was elevated, and tubulin disruption took place in both CCRF-CEM (non-chemoresistant) and CEM/C1 (chemoresistant) cells treated with MBZ for 24 h. Alongside caspase activation and tubulin disruption, Notch1 signaling (typically activated in T-ALL cells) was inhibited by MBZ in western blot experiments. Furthermore, in vivo experiments using T-ALL murine models showed a significant decrease of lymphocytes and CD4+ cells in MBZ-treated T-ALL mice, further indicating that MBZ may be developed as a therapeutic agent for chemoresistant T-ALL cells in vitro and in vivo [40].

### 4.3. Multiple Myeloma (MM)

MM is a cancer of plasma cells in the bone marrow [50]. It can be treated with alkylating agents, corticosteroids, immunomodulatory drugs, and proteasome inhibitors [51]. However, MM is characterized by multiple remissions and recurrences. Studies have shown that with each consecutive treatment, the duration of response decreases due to the development of acquired drug resistance [52,53]. Anthelmintic benzimidazoles have been found to be effective against MM, as discussed below.

#### 4.3.1. Mebendazole

A common oncogenic event occurring in MM is the upregulation of c-Maf, an oncogenic transcription factor [54]. In a study by Hurt et al., suppression of c-Maf was shown to restrict myeloma proliferation [54]. Treatment with MBZ demonstrated cytotoxicity and synergized with ionizing radiations, various chemotherapeutic agents, and stimulated antitumoral immune responses in cancer cells [55]. Furthermore, Chen et al. observed synergistically increased apoptosis in MM cells treated with MBZ in combination with other anticancer drugs, such as daunorubicin or WP1130 (USP5 inhibitor) [41]. MBZ was also found to exhibit antimyeloma activity by suppressing USP5, a deubiquitinase, leading to the degradation of c-Maf. Chen et al. also tested MBZ’s in vivo efficacy using a human MM xenograft mouse model, where LP1 or RPMI-8226 cells were subcutaneously injected into nude mice. Mice treated with MBZ (50 and 100 mg/kg) for 20 days exhibited a dose-dependent tumor suppression without signs of toxicity. Tumor tissues from the MBZ-treated mice showed elevated levels of apoptosis markers, including cleaved caspase-3 and PARP. This study concluded that MBZ disrupts the USP5/c-Maf axis, leading to apoptosis in MM cells [41].

#### 4.3.2. Albendazole

Based on past research, specifically conducted by Yi et al. and Kim et al., it can be deduced that ABZ has antiproliferative effects on MM cell lines [29,30]. Both studies established the therapeutic potential of ABZ as well as its combined impact with bortezomib, a proteasome inhibitor and a treatment drug for MM [56]. According to Yi et al., ABZ is capable of resensitizing cells that are resistant to bortezomib by inhibiting the nuclear factor kappa-B (NF-κB) pathway and reducing the number of aldehyde dehydrogenase (ALDH)-positive multiple myeloma stem cell-like cells (MMSCs), which commonly cause resistance to drugs [29]. Furthermore, Kim et al. demonstrated a synergistic G2/M phase arrest and induction of apoptosis using ABZ in combination with bortezomib [30].

#### 4.3.3. Flubendazole

FLBZ is one of the other anthelmintic agents that exhibited promising antimyeloma activity [34]. MM patients respond to conventional chemotherapy but eventually develop resistance to different treatments (anthracyclines, alkylating agents, proteasome inhibitors, and immunomodulators, etc.) due to upregulation of efflux transporters such as P-glycoprotein (P-gp) [57]. Vinca alkaloids (vinblastine, vincristine) are widely used agents for the treatment of adults with MM, and increased P-gp activity results in resistance to alkaloids [58]. The findings from Spagnuolo et al. showed that FLBZ effectively induces cell death at nanomolar concentration in different MM cell lines (OPM2, KMS11, JJN3 LP1, H929, L1, KMS12, KSM18, and OCI My5) through inhibition of tubulin polymerization distinct from that of Vinca alkaloids [34]. The activity of FLBZ is not affected in vinblastine-resistant cells that overexpress P-gp. Furthermore, FLBZ (50 mg/kg; i.p.) exhibited antimyeloma activity by significantly delaying tumor growth in mice xenografts injected subcutaneously with OPM2 MM cells. Interestingly, the combination of FLBZ with Vinca alkaloids exhibited synergistic cytotoxicity in MM cells both in vitro and in vivo [34]. Altogether, FLBZ is a potent microtubule inhibitor that could be used alone or in combination with Vinca alkaloids for the treatment of myeloma.

### 4.4. Chronic Myeloid Leukemia (CML)

CML is a build-up of granulocytes in the blood and bone marrow. Imatinib is an FDA-approved drug commonly used for CML treatment [59]. Daniel et al. point to a T315I gate-keeper mutation in ABL1 proteins, which are constitutively active in CML, that cause resistance to some drugs [42]. Combining the use of MBZ and tyrosine kinase inhibitors (TKIs) such as dasatinib and imatinib can overcome this drug resistance by disrupting the cell cycle in both sensitive and resistant K562 cell lines [42]. Other studies supported MBZ’s ability to induce cell death of CML. Hu et al. revealed that MicroRNA-150-5p (miR-150-5p), which aids in hematopoietic development, is associated with MYB expression, a proto-oncogene transcription factor, in CML, which is linked to hematopoietic stem cell differentiation [60]. MBZ inactivates MYB, thereby reducing cell cycle progression and inducing apoptosis [60]. In addition, it was observed that decreased levels of miR-150 play a role in resistance to tyrosine kinase inhibitors in CML cells [61]. The association between MBZ and tyrosine inhibitors, miR-150, and MYB suggests MBZ is a potential anti-CML treatment.

### 4.5. Lymphoma

Lymphomas generally originate from lymphocytes (B-cells and T-cells/natural killer cells) and are diagnosed based on the cytogenetic, molecular, and clinical level characteristics along with morphology and immunophenotype analysis of hematopoietic cells [62]. The lymphoid malignancies account for 5% among all types of cancer and cause deaths globally due to lack of effective treatments, drug resistance, and relapse. Therefore, there is a desperate need for the development of new agents and treatment approaches for lymphoma treatment alone or in combination with standard care therapy [63]. Drug repurposing offers a feasible and reliable approach to identifying new small molecules that can inhibit lymphoma progression. Several traditionally anthelmintic benzimidazoles have been shown to be potent anticancer agents that significantly inhibited lymphoma progression during repurposing screens. This is discussed as follows:

In 2008, for the first time, Gao et al. serendipitously identified the anticancer activity of FBZ when combined with dietary supplementation of vitamins [32]. The CB17/Icr-*Prkdc^scid^*/IcrIcoCrl (SCID) mice failed to grow an established human lymphoma xenograft when FBZ was used to treat rodent pinworm infections. Further investigation in 4-week-old SCID mice showed that the group pretreated for 2 weeks with a combination of both FBZ and vitamins had significantly reduced tumor growth compared to mice fed with the standard diet, FBZ alone, or vitamins alone [32]. Elsewhere, a case report by Abughanimeh et al. indicated that the use of FBZ in an 83-year-old male diagnosed with Diffuse Large B-Cell Lymphoma (DLBCL) who refused to take chemotherapy resulted in disease regression as his PET/CT scan revealed interval improvement in disease with no new lesion [33].

In 2015, Martin Michaelis screened FLBZ in a panel of 321 cell lines, including cell lines from 26 cancer entities [35]. Multiple myeloma, neuroblastoma, and leukemia/lymphoma consistently belonged to the cancer entities that displayed the highest sensitivity to FLBZ. In leukemia/lymphoma, 40 out of 49 cell lines (82%), in multiple myeloma, 10 out of 10 cell lines (100%), and in neuroblastoma, 28 out of 32 cell lines (88%) displayed an IC90 < 1 μM. Together, these three entities accounted for 78 of the 117 cell lines (67%) that displayed IC90’s < 1 μM among the 26 cancer entities [35].

Carpenter et al. have reported benzimidazole derivatives as integrin α4β1 antagonists that are potent for lymphoma treatment [64]. These authors reported the rapid microwave preparation, structure-activity relationships, and biological evaluation of medicinally pertinent benzimidazole heterocycles as integrin α4β1 antagonists. They also documented tumor uptake of ^125^I-labeled derivatives in xenograft murine models of B-cell lymphoma. Molecular homology models of integrin α4β1 predicted that docked halobenzimidazole carboxamides have the halogen atom in a suitable orientation for halogen-hydrogen bonding. The high-affinity halogenated ligands identified offer attractive tools for medicinal and biological use, including fluoro and iodo derivatives with potential radiodiagnostic (^18^F) or radiotherapeutic (^131^I) applications, or chloro and bromo analogues that could provide structural insights into integrin-ligand interactions through photoaffinity, cross-linking/mass spectroscopy, and X-ray crystallographic studies [64].

Muhammad et al. established the anti-lymphoma activity of benzimidazole-substituted zinc phthalocyanine derivatives in B-cell lymphoma cell lines and established their cytotoxicity mechanism through physical interaction with DNA, including minor groove binding and intercalation between bases [65]. Taken together, benzimidazoles and their derivatives could act as potential anticancer agents and supplement current lymphoma treatments for effective regression of lymphoid malignancies. Further studies should be directed to evaluate these anthelmintic benzimidazoles’ efficacy in combination with other standard therapy agents.

## 5. Mechanism of Action of Benzimidazoles against Hematological Cancers

The repurposing of anthelmintic benzimidazoles as anticancer agents has demonstrated significant antileukemic activity across various hematological malignancies, as discussed in earlier sections. Figure 4 summarizes the different mechanisms through which benzimidazole-based derivatives exert their antileukemic effects, with further details provided below.

### 5.1. Tubulin Depolymerization

Microtubules play a vital role in mitosis, cell signaling, and motility. Numerous agents that disrupt microtubules have been developed as anticancer agents [66,67]. Benzimidazole carbamates are broad-spectrum anthelmintic drugs that inhibit tubulin polymerization in parasite cells and consequently were first tested for anticancer activity in 2002 [68]. MBZ exhibited anticancer activity in Non-Small Cell Lung Cancer (NSCLC) cells by disrupting spindle formation along with induction of tubulin depolymerization [68]. Later, MBZ was shown to inhibit tubulin polymerization in glioblastoma cell lines and showed antiproliferative efficacy in combination with temozolomide [69]. MBZ was also found to potently inhibit migration and invasion of gastric cancer cell lines from patients by disrupting microtubule structure. Importantly, MBZ shows more efficient cytotoxic activity than other clinically approved chemotherapeutic agents such as 5-fluorouracil, cisplatin, oxaliplatin, irinotecan, gemcitabine, paclitaxel, and doxorubicin in gastric cancer cells [70].

Due to inhibition of tubulin formation and favorable pharmacokinetic and toxicity profiles, MBZ was proposed as a replacement for the anti-tubulin agent vincristine for the treatment of brain tumors [71]. In addition to MBZ, FBZ is also shown to inhibit tubulin structure, polymerization, and function in PPC-1 prostate cancer cells [34]. Treatment of OCI-AML2 leukemic cells with FBZ resulted in mitotic catastrophe and chromosomal missegregation enumerated by an increased number of multinucleated cells, similar to other microtubule inhibitors. Further mechanistic binding studies using purified bovine tubulin in the presence of colchicine indicated that FBZ interacts with tubulin through a mechanism distinct from vinblastine. Due to the differences in tubulin binding site, FBZ acted synergistically with vinka alkaloids (vincristine, vinblastine) to inhibit leukemia progression in vivo [34].

Recently, Will Castro et al. have shown that treatment with ABZ results in chromosomal segregation defects due to inhibitory effects on the mammalian spindle apparatus. A very high frequency of aneuploid cells was observed after treatment with lower doses of ABZ. ABZ was found to be 60-fold more potent compared to Noscapine, a known spindle poison that affects microtubule dynamics [72]. Taken together, it is evident that these anthelmintic benzimidazoles inhibit tubulin polymerization (Figure 4A) in different cancer cells and can be repurposed as an effective antiproliferative agent.

### 5.2. Cell Cycle

A tight regulation of the cell cycle is required for the maintenance of cellular homeostasis. Genetic mutations or aberrant regulation of the cell cycle result in malignant transformation. Cancer prevention could be achieved through the strict regulators of the cell cycle, which ultimately define the cellular fate [73,74]. In several instances, benzimidazole-based derivatives have been shown to exhibit anticancer effects through regulation of cell cycle progression. The use of MBZ in K562 by itself and with imatinib decreased cells in the G0 and G1 phases of the cell cycle and elevated DNA fragmentations (Sub-G1 phase). In addition, combined use of MBZ with other TKI inhibitors such as dasatinib significantly decreased the number of cells in the S-phase. Likewise, in the FEPS cell line, the surge in DNA fragmentation was statistically different upon combined treatment with MBZ and TKI inhibitors (imatinib, dasatinib) compared to single agents [42].

In 2018, Wang et al. reported that ABZ treatment resulted in G2/M cell cycle arrest and downregulation of Sirtuin-3 (SIRT3) in U937 leukemia cells. The hypodiploid cells exhibited sub-G1 phase arrest compared to untreated control cells after ABZ treatment. The effects of ABZ on cell cycle were neutralized upon overexpression of SIRT3 and inferred the contribution of microtubule-destabilizing activity [75]. Furthermore, a recent study by Will Castro et al. demonstrated that the treatment of DNA damage-tolerant (DDT) lymphoma cells with ABZ led to the G2/M cell cycle arrest and increased sub-G1 cells in a dose-dependent fashion [72]. The effects on the G2/M cell cycle phase in leukemic cells with ABZ were consistent with other types of cancer cells [76,77]. Therefore, the effect of anthelmintic benzimidazoles on cell cycle plays a critical role in mediating anticancer effects in blood cancer cells (Figure 4B).

### 5.3. Loss of Heterozygosity

Loss of heterozygosity (LOH) is a common type of genetic mutation in several types of cancer that results in the allelic imbalance by which heterozygous somatic cells become homozygous because one copy of the two alleles gets lost [78]. LOH is usually caused by either degradation, deletion due to an imbalance rearrangement, gene conversion, mitotic recombination, or loss of a whole chromosome. LOH creates genetic differences between tumor and normal cells and provides an approach to selectively target cancer cells and biomarkers for cancer risk assessment [79,80].

In 2021, Will Castro et al. reported that ABZ is a potent inducer of LOH and accelerator of chromosomal missegregation in mammalian cells [72]. The transcriptome of cells exposed to ABZ revealed a cell cycle signature supporting the mitosis defects. Treatment of the DDT lymphoma cell line with ABZ exhibited a dose-dependent increase in the aneuploid cells as characterized by the subG1 and subG2 cells in cell cycle analysis. Focused cellular microscopic analysis implicated ABZ as a potent spindle poison due to its effect on the human spindle apparatus and microtubule dynamics. The authors have assessed chromosomal missegregation induced by ABZ, taking advantage of an in vitro approach that enables direct measurement of the LOH frequency [81]. Briefly, embryonic stem (ES) cells that are haploinsufficient for the mismatch repair (MMR) gene Mlh1 were exposed to ABZ for 24 h, and then LOH cells were selected after 2-week exposure to 6-thioguanine (6-TG), which selectively kills MMR proficient cells. The LOH cells that are MMR deficient due to loss of remaining functional Mlh1 allele after ABZ treatment survived in the presence of 6-TG. Furthermore, LOH induced by ABZ was assessed in vivo using Msh2 heterozygous (Msh2+/−) mice, a model for Lynch Syndrome (LS). MMR-deficient crypt foci, which later progress into tumors, are commonly found in the intestinal tract of LS patients [82]. Immunohistochemical staining of the small intestinal tract of the mice treated with ABZ for 32 consecutive days showed a significant increase in the number of MSH2-deficient crypts and thus indicated that ABZ induces LOH in vivo. Since chromosomal segregation defects and LOH in haploinsufficient carriers of tumor suppressor genes accelerate cancer, ABZ could predispose to tumor development besides its general cytotoxicity (Figure 4C). These data conclude that ABZ acts as a spindle poison and induces LOH and chromosomal segregation defects in mammalian cells [72]. Further mechanistic studies are necessary to understand the effects of other anthelmintic benzimidazoles prior to their repurposing as anticancer agents.

### 5.4. Differentiation Therapy

Differentiation therapy has several advantages over standard chemotherapy in that differentiation therapy illustrates irreversible effects and prompt reduction of tumor burden through terminal maturation of blast cells. One notable example of this includes the combination of all-trans retinoic acid (ATRA) and arsenic as the treatment option for acute promyelocytic leukemia (APL) [22,23]. KalantarMotamedi et al. performed cheminformatics analysis to identify novel small molecules that can induce differentiation through transcriptional drug repurposing [31]. Their approach was validated by the identification of retinoic acid as one of their hits. They further identified fenbendazole as a novel potential agent (scored 20) with a scope for repurposed as an antileukemic agent in both bioinformatics (in the top 30) and cheminformatics. They further validated the chemoinformatic results in vitro, obtaining LC_50_ values of 0.50 μM, 0.36 μM, and 0.31 μM at 24, 48, and 72 h using HL-60 cells. Further testing in human bone marrow stem cells concluded that fenbendazole exhibited a 14.5-fold selectivity of cancer cells over somatic cells. Giemsa staining of HL-60 cells after FBZ showed the presence of granulocytes and induction of apoptosis through terminal differentiation [31].

Li et al. developed a novel computational approach to define or detect the differentiation state of hematopoietic malignancies using the Lineage Maturation Index (LMI). This novel approach can detect known changes of differentiation state in both normal and malignant hematopoietic cells. They tested a small molecule library in HL-60 leukemic cells and found that mebendazole exhibited a high LMI score. MBZ was found to greatly induce differentiation of leukemic blast cells based on changes in cell surface markers (CD11b, CD11c, and CD14) and cell morphology in terms of increased chromatin condensation, decreased nuclear-to-cytoplasmic ratio, reduction in cytoplasmic basophilia, and accumulation of granules in the cytoplasm [37].

Noura et al. performed a chemical screening and identified ABZ as a promising compound that can induce differentiation of non-APL AML cells via the Krüppel-like factor 4-dihydropyrimidinase-like 2A (KLF4-DPYSL2A) differentiation axis to monocytes [28]. Therefore, these benzimidazoles could serve as an effective and safe agent to induce terminal differentiation for the treatment of hematological malignancies associated with the accumulation of abnormal blast cells (Figure 4D).

### 5.5. Proteasomal Degradation

The proteasome, a multicatalytic proteinase complex, plays a critical role in cell survival through precise control of intracellular protein turnover. The ubiquitin-proteasome pathway accounts majorly for protein degradation in eukaryotes. The refinement and targeting of proteasome activity render cancer cells to undergo apoptosis, and this approach provides an effective strategy for cancer treatment and prevention [83,84]. Anthelmintic benzimidazoles have been shown to modulate proteasome function to exert anticancer activity as described in the following section (Figure 4E).

#### 5.5.1. c-MYB Degradation

Genetic rearrangements in the MLL gene result in MLL-fusion proteins that are frequently associated with the development of blood cancers such as AML. c-MYB is an important transcription factor that drives the expression of several oncogenes, including MLL-fusion genes, thus contributing to the development and progression of AML [85]. c-MYB is critical for normal hematopoiesis and maintenance of stem cell self-renewal and has been associated with hematological cancers [86,87,88]. Leukemia cells rather than normal hematopoietic cells have been shown to be sensitive to inhibition of c-MYB. Walf-Vorderwülbecke et al. have profiled c-MYB gene expression in MLL-rearranged AML and identified anthelmintic MBZ as a top hit to target c-MYB through screening of the Connectivity Map database [38]. Further in vitro testing revealed inhibitory effects of MBZ on the proliferation and clonogenicity of both MLL-rearranged and non-rearranged human AML cell lines (THP-1, OCI-AML3, NOMO-1, KCL22, U937, MV4-11, KASUMI-1, and SHI-1). Similar effects were observed with siRNA-mediated knockdown of c-MYB in those AML cell lines. In MBZ-treated AML cells, c-MYB undergoes proteasomal degradation by interfering with the heat shock protein 70 (HSP70) chaperone system without affecting c-MYB at the RNA level. In addition, authors have determined that MBZ-targeted c-MYB degradation is independent of its microtubule depolymerization activity. In line with these observations, THP1 transplanted NSG mice show reduced leukemia progression in vivo with oral administration of MBZ. To summarize, MBZ induces proteasomal degradation of the c-MYB transcriptional factor while impairing AML progression. The c-MYB degradation represents a novel approach to selectively target leukemic cells [38].

#### 5.5.2. Degradation of GLI Transcription Factors

A highly conserved Hedgehog (HH) signaling pathway plays a crucial role in several cellular functions, including but not limited to cell differentiation and stem cell maintenance [89]. GLI transcription factors (TFs) are the main effectors of the HH signaling pathway that drive a transcriptional program to promote growth, migration, survival, and stemness. In addition to the canonical HH pathway, various signaling cascades such as TGF-β, PKC, FLT3/STAT5, RAS/RAF/MEK/ERK, and PI3K/AKT/mTOR result in non-canonical activation of the GLI TFs (GLI1/2/3) [90]. Aberrant expression of GLI TFs is associated with a poor prognosis in a wide variety of cancers and leads to overactivation of several tightly regulated cellular processes such as proliferation, angiogenesis, and Epithelial-to-mesenchymal transition (EMT) [39,45]. In AML, high GLI1/2 expression indicates a negative prognostic marker, and targeted inhibition intervenes in antileukemic effects in vitro and in vivo [91]. Targeting GLI TFs has recently become a major focus of potential therapeutic protocols. Moreover, GLI transcription factors play a fundamental role in the maintenance of leukemia-initiating cells that are responsible for therapy failure and tumor relapse due to their inherent chemotherapy resistance [92].

MBZ induced strong and dose-dependent anti-leukemic effects through sensitization of AML cells to standard chemotherapy with cytarabine [39]. Treatment of AML cells with MBZ resulted in downregulation of intracellular protein levels of GLI1/GLI2 transcription factors and reduced the GLI promoter activity in luciferase-based reporter assays. Further mechanistic analysis revealed that MBZ promotes the proteasomal degradation of GLI transcription factors through inhibition of HSP70/90 chaperone activity to mediate its anti-leukemic effects. Additional molecular dynamics simulations performed on the MBZ-HSP90 complex showed a stable binding interaction at the ATP binding site. Interestingly, one out of two patients with refractory AML treated with MBZ showed a significant reduction in peripheral blood blast count along with GLI signaling activity in a plasma assay. Therefore, MBZ is an effective GLI inhibitor that has the clinical potential to be evaluated alone or in combination with standard conventional chemotherapy for AML treatment [39].

### 5.6. Signaling Pathways

#### 5.6.1. Phosphoinositide 3-Kinase (PI3K)/AKT Signaling

Phosphoinositide 3-kinase (PI3K)/AKT signaling plays a vital role in cell cycle progression and oncogenic transformation [93]. In addition, gain-of-function mutations of the MAPK/ERK pathway are associated with most cancers [94]. Pharmacological inhibition of PI3K/Akt and MAPK/ERK pathways has gained attention in recent years as anticancer approaches [95,96,97]. Recently, He et al. showed that MBZ treatment abrogated the activation of both Akt and Erk1/2 in AML leukemic cells [36]. Treatment of U937 leukemic cells for 24 h with MBZ exhibited a dose-dependent reduction in the phosphorylated Akt and Erk1/2 proteins without affecting total Akt and Erk1/2 levels. MBZ suppressed the growth of leukemic cells in vivo and prolonged the survival in an AML xenograft mouse model [36]. Therefore, it was proposed that the antileukemic effects of MBZ were partly mediated by inhibition of PI3K/Akt and Erk1/2 signaling pathways (Figure 4F).

#### 5.6.2. Nuclear Factor Kappa B (NF-κB) Pathway

Nuclear factor kappa B (NF-κB) is a nuclear transcription factor that regulates cell survival, autophagy, angiogenesis, and cell death [98]. Aberrant NF-kB activation has been linked to malignant transformation, cancer cell progression, and drug resistance [99]. Selective inhibition of NF-kB signaling by natural compounds, nutraceuticals, and chemical agents has been shown to suppress the proliferation of cancer cells and has emerged as a potential molecular target for cancer treatment [63,99,100,101]. In a recent study, the anthelmintic benzimidazole ABZ was shown to inhibit proliferation of MM cells in vitro and in vivo through suppression of the NF-κB pathway [29]. RNA-Seq analysis after ABZ treatment of ARP1 MM cells exhibited differential regulation and enrichment of NF-kB pathway genes. Also, western blot analysis of MM cell lines (ARP1 and ANBL6) after ABZ indicated a dose-dependent decrease in the levels of p65 and phospho-p65, an important transcription factor component of NF-κB. Since elevated activity of NF-kB was reported in cancer stem cells, treatment of MM cells with ABZ resulted in enhanced cytotoxicity in ALDH1+ve MMSCs, a rare subpopulation of MM cells exhibiting self-renewal and multidrug resistance. ABZ treatment re-sensitized cells resistant to bortezomib (BTZ) and overcome bortezomib resistance by decreasing ALDH1+ MMSCs numbers. Moreover, ABZ exhibited synergetic activity in combination with SC75741, a specific inhibitor of the NF-κB pathway [102]. The tumor tissues from mice treated with ABZ showed a drastic decrease in the levels of proliferation markers Ki67, p65, and ALDH1A in immunofluorescence, immunohistochemical, and immunoblotting analysis. ABZ treatment inhibited NF-kB signaling, which is activated in ANBL6-BR (resistant to BTZ) cells [29]. Altogether, it was concluded that ABZ overcomes the BTZ resistance and targets cancer stem cells through inhibition of the NF-kB signaling pathway in MM (Figure 4F).

#### 5.6.3. Notch1 Signaling

Notch signaling is critical for the homeostasis of hematopoietic stem cells (HSCs) and T cell lineage commitment [103]. The mutations in Notch1 result in the aberrant transcriptional activation of its downstream genes (MYC, HES1, HEY, CCND1, etc.) involved in anabolic pathways and thus drive the neoplastic progression of several types of cancers [104,105]. Notch1 is tightly connected to other oncogenic signaling pathways and impacts proliferation, apoptosis, chemosensitivity, immune response, and cancer stem cells. Notch1 inhibition via either γ-secretase inhibitor, short interference RNA, or monoclonal antibodies offers a great approach alone or in combination with other therapeutic agents to combat cancer [104,106].

It was reported by Wang et al. that anthelmintic MBZ suppresses proliferation of T-ALL cells through inhibition of Notch1 signaling [40]. T-ALL cell lines CCRF-CEM and CEM/C1 treated with MBZ for 24 h showed reduced levels of Notch1 and its downstream transcriptional factors c-Myc and Hes1 in immunoblotting analysis. Interestingly, the expression levels of γ-secretase that catalyzes proteolytic cleavage of the Notch1 receptor were not altered by MBZ treatment of T-ALL cell lines. It was also found out that MBZ inhibited the Notch1 intracellular domain (NICD) in CEM/C1 cells. The anticancer effects of MBZ were investigated in vivo using a leukemia model generated from BALB/c nu/nu mice by intravenously injecting chemoresistant CEM/C1 cells [107]. Expression analysis by RT-PCR in the T-ALL mice spleens showed activation of Notch1 signaling, and MBZ treatment (100 mg/kg/d; i.p.) for 10 continuous days reduced the mRNA levels of Notch1, c-Myc, and Hes1. Therefore, MBZ inhibits Notch1 signaling to exert antileukemic activity in vivo [40]. Altogether, it was concluded that MBZ suppresses cancer progression through inhibition of the Notch1 signaling pathway, and Notch1 could be a critical target in T-ALL (Figure 4F). Further studies are needed to understand the effect of MBZ and other benzimidazoles on the oncogenic signaling pathways to mediate anticancer effects.

## 6. Conclusions and Future Perspectives

In conclusion, this comprehensive exploration of anthelmintic benzimidazole derivatives, specifically mebendazole, fenbendazole, and albendazole, highlights their versatile and promising roles as potential therapeutic agents against hematological cancers, with a specific emphasis on leukemia. These classes of drugs demonstrate a range of mechanisms encompassing microtubule disruption, cell cycle regulation, induction of differentiation, and modulation of crucial signaling pathways such as PI3K/AKT, NF-κB, and Notch1. The observed effects on tubulin depolymerization, cell cycle arrest, and proteasomal degradation of key transcription factors contribute to their anti-leukemic properties. As stated above, several ongoing clinical trials are aimed at evaluating the anticancer potential of benzimidazole-based agents against different malignancies, including blood cancers. Future research avenues should delve deeper into the intricate molecular pathways impacted by these benzimidazoles, optimize their therapeutic efficacy, and explore potential synergies with existing standard treatments. As most conventional benzimidazole-based derivatives suffer from poor aqueous solubility, rapid clearance, and toxicity (teratogenicity and embryotoxicity), there is a need for novel derivatives and formulations to enhance the safety, stability, and on-target efficacy. This research findings summarized in this review lay a robust foundation for ongoing investigations, aiming to translate these discoveries into innovative and enhanced strategies for treating hematological malignancies, improving patient outcomes through repurposing of anthelmintic benzimidazoles.

## Figures and Tables

**Figure 1 cancers-16-03454-f001:**
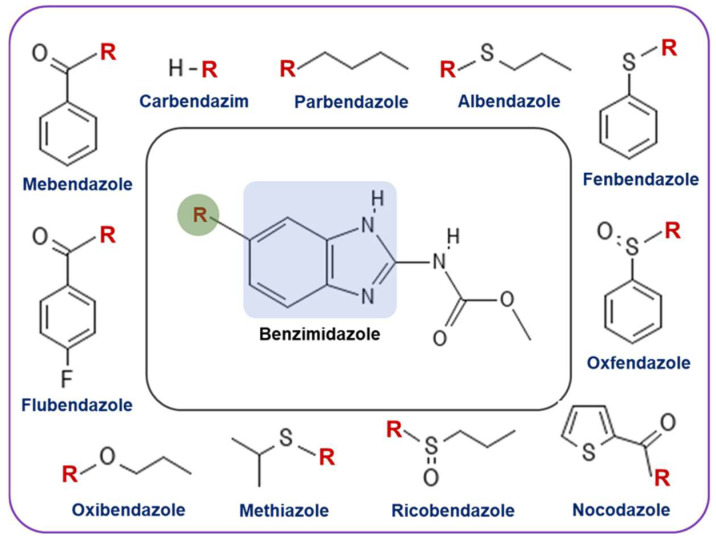
Structure of various benzimidazole derivatives that are approved by the FDA as anthelmintic agents. Among these, parbendazole, fenbendazole, oxfendazole, and oxibendazole are used to treat animals, while albendazole, mebendazole, ricobendazole, and flubendazole are used for fighting parasite infections in humans.

**Figure 2 cancers-16-03454-f002:**
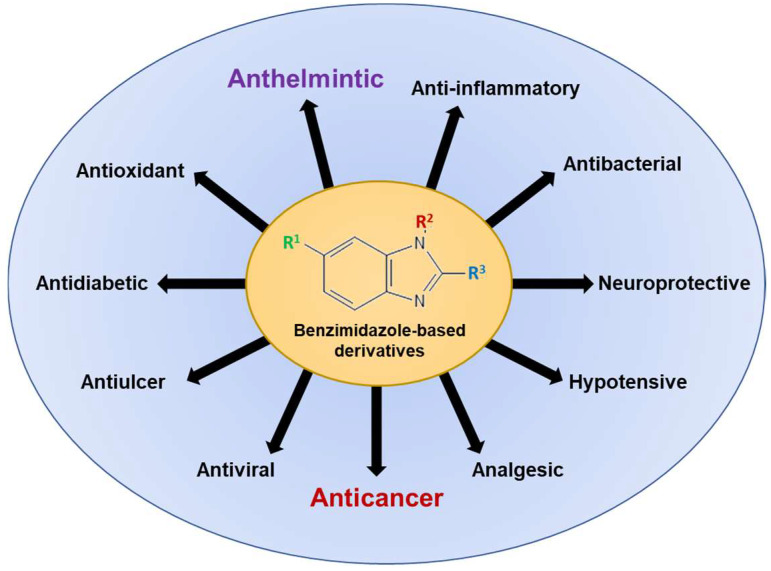
The spectrum of biological activities associated with benzimidazole-based derivatives for human use. Conventionally, benzimidazoles were approved as anthelmintic/antiparasitic agents, and later several other derivatives were developed. Currently, the exploration of benzimidazoles and their derivatives for their anticancer activity is of great interest to researchers.

**Figure 3 cancers-16-03454-f003:**
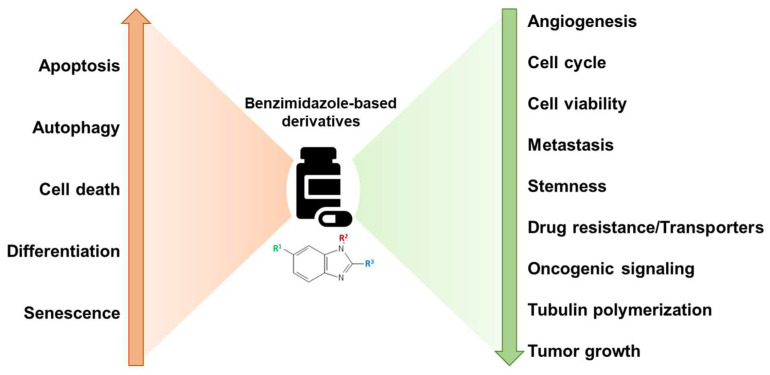
Broad mechanisms of action of benzimidazoles as anticancer agents. Benzimidazole derivatives inhibit cell cycle, invasion/metastasis, tubulin polymerization, oncogenic signaling, angiogenesis, and cell viability, along with increasing apoptosis, autophagy, and terminal differentiation for reducing the cancer progression.

**Figure 4 cancers-16-03454-f004:**
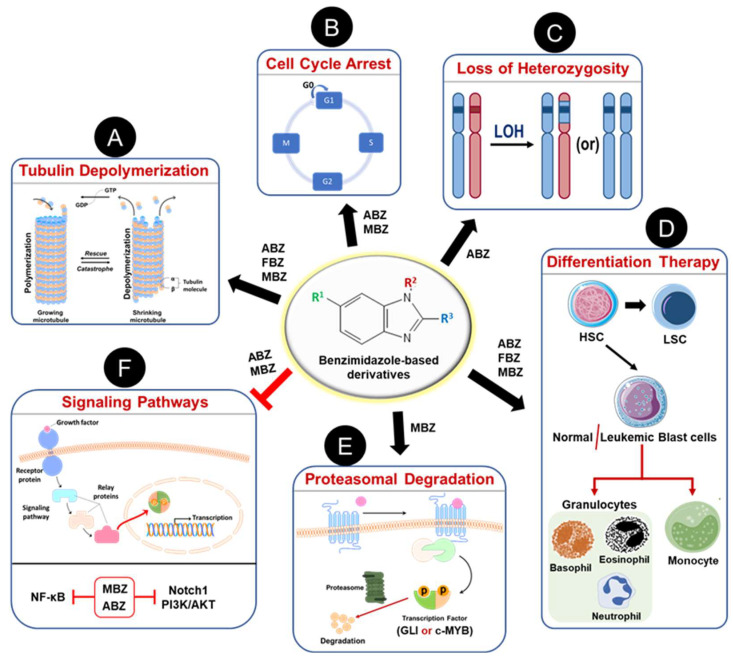
Mechanisms of action of anthelmintic benzimidazole-based derivatives as anti-leukemic agents. As a classical anthelmintic, BMDZs inhibit tubulin polymerization (**A**) to induce cell death by mitotic cell arrest (**B**). Albendazole (ABZ) known to accelerate chromosomal missegregation and thus induces loss of heterozygosity in mammalian cells (**C**). Some BMDZs are known to induce terminal differentiation of leukemic blast cells to granulocytes and/or monocytes and thus cause cell death (**D**). Treatment with BMDZs promotes proteasomal degradation of transcriptional factors such as GLI and c-Myb, which play a vital role in leukemia development and progression (**E**). Also, these BMDZs inhibit several oncogenic signaling pathways (Notch1, PI3K/AKT, NF-κB) to achieve remission during the treatment of hematological malignancies (**F**).

**Table 1 cancers-16-03454-t001:** Summary of experimental evidence (in vitro and in vivo) that described drug repurposing of anthelmintic benzimidazoles for the treatment of hematological cancers.

Benzimidazole	Blood Cancer	In Vitro (or) In Vivo Models	Observations	References
Albendazole(ABZ)	AML	⮚ THP-1 and KO52 cell lines ⮚ Primary murine cells with MLL-AF9 translocation and AML cells with normal karyotype	⮚ Reduced cell viability ⮚ Inhibited colony formation ⮚ Negligible effect on normal bone marrow cells ⮚ Induced terminal differentiation of non-APL AML cells	[18,28]
	MM	⮚ ANBL6, ARP1, KMS11, and MM.1S, MM.1R, RPMI8226, and U266 cell lines	⮚ Reduced cell growth and viability ⮚ Synergistic induction of apoptosis and G2/M arrest in combination with bortezomib ⮚ Resensitized bortezomib-resistant cells ⮚ Inhibited NF-κB pathway ⮚ Reduced the number of ALDH-positive MMSCs	[29,30]
Fenbendazole(FBZ)	AML	⮚ HL60 cell line	⮚ Reduced cell viability ⮚ Negligible effect on the normal bone marrow cells ⮚ Induced terminal differentiation of HL60 cells to granulocytes	[31]
	Lymphoma	⮚ C.B-17/Icr-*prkdc^scid^*/Crl (SCID) mice	⮚ Reduced tumor growth in mice on diet supplemented with vitamins	[32]
		⮚ 83-year-old male with Diffuse Large B-cell lymphoma	⮚ Observed tumor regression with repeated PET/CT scan ⮚ Found safe with no new lesions	[33]
Flubendazole(FLBZ)	AML	⮚ U937, MDAY, NB4, OCI-AML2 cell lines ⮚ Primary AML patient samples ⮚ SCID mice injected subcutaneously with OCI-AML2 cells	⮚ Reduced cell viability ⮚ Inhibited colony formation ⮚ Arrested cells in G2 phase ⮚ Exhibited synergy with Vinka alkaloids. ⮚ Reduced tumor growth in mice	[34]
	MM	⮚ MM cell lines (OPM2, KMS11, JJN3 LP1, H929, L1, KMS12, KSM18, and OCI My5) ⮚ SCID mice injected subcutaneously with OPM2 cells	⮚ Reduced cell viability ⮚ Reduced tumor growth in mice	[34]
	Lymphoma	⮚ Cell lines	⮚ Reduced cell viability in 80% tested cells ⮚ Cells displayed an IC90 < 1 μM	[35]
Mebendazole(MBZ)	AML	⮚ AML cell lines (MV4-11, MOLM-13, OCI-AML3, HL-60, THP-1, and U937) ⮚ Primary AML patient samples ⮚ THP-1 xenograft mice model	⮚ Reduced cell proliferation ⮚ Inhibited colony formation ⮚ No effect on normal CD34+ cells ⮚ Inhibited protein folding through the HSP70 chaperon system ⮚ Induced terminal differentiation ⮚ Resulted in the degradation of transcription facts (c-MYB and GLI) ⮚ Impaired leukemia progression and extended survival of mice	[36,37,38,39]
	ALL	⮚ T-ALL cell lines (CCRF-CEM and its chemoresistant derivative, CEM/C1) ⮚ T-ALL mice inoculated with CEM/C1 cells	⮚ Reduced cell proliferation ⮚ Decreased cell viability ⮚ Treated mice showed a reduction in lymphocytes and CD4+ T-cells	[40]
	MM	⮚ MM cell lines (RPMI-8226, LP1, and U266) ⮚ Myeloma xenograft mice injected with LP1 and RPMI-8226 cells	⮚ Induced apoptosis ⮚ Increased c-Maf degradation ⮚ Induced proteasomal degradation ⮚ Synergizes with WP1130, a USP5 (deubiquitinase) inhibitor ⮚ Delayed tumor growth in MM xenograft mice	[41]
	CML	⮚ CML cell lines (K562 and its chemoresistant derivative FEPS)	⮚ Increased cytotoxicity ⮚ Cell cycle arrest ⮚ Induced apoptosis ⮚ Synergizes with tyrosine kinase inhibitors such as imatinib and dasatinib	[42]

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
