# Peer review of "From Deworming to Cancer Therapy: Benzimidazoles in Hematological Malignancies"

_cancers, 2024, doi:10.3390/cancers16203454_

Round 1
Reviewer 1 Report
Comments and Suggestions for Authors
1) They organized the review in the order of 1. Introduction, 2. Benzimidazoles as Anthelmintic Agents, 3. Repurposing of Anthelmintic Benzimidazoles as Anticancer Agents, 4. Repurposing of Anthelmintic Benzimidazoles for the Treatment of Hematological Cancers, etc.
It is recommended that the structure of some parts of this review be reorganized.
In section 3 (Repurposing of Anthelmintic Benzimidazoles as Anticancer Agents: they mentioned benzimidazoles for other uses as follows: “Various other benzimidazole derivatives are under clinical development for other biological activities, such as anti-inflammation and anticoagulation (lines 177).” It is not appropriate for this section. It needs to be modified.
In section 4, to help readers understand and make To focus on the aim of this review, it needs to categorize the types of hematological cancers, describe the pathology that is problematic in each type, and then mention about the benzimidazoles that act on them and show anticancer effects.
In addition, in section 4, they described the anticancer effect of benzimidazoles on other types of cancer, such as head and neck squamous cell carcinoma, glioma, melanoma, and colorectal cancer (lines 318~320). There are various reports describing the anticancer effect of benzimidazoles on another cancer type. It is not proper in this section.
2) In addition, Figure 4 needs to be updated.
It is recommended to include the contents, which were introduced in the section 5 (Mechanism of Action of Benzimidazoles against Hematological Cancers) and to include what kind of benzimidazole derivative acts on each mechanism and how they affect (inhibit or activate).
3) It would be better to summarize the contents described in section 4 of Table as recommended purpose of Anthelmintic Benzimidazoles.
4) Abbreviations should be used after defining the full name of some words, and then keep using abbreviations. However, they mix the full names and abbreviations, and sometimes they define some words again.
5) It needs to include references for Figure 3 (lines 121~123).
And please check the reference lists. For example, #11. There is no source of the article.
Author Response
Responses to Reviewer-1 Comments:
Response: We sincerely appreciate the reviewer for taking the time to thoroughly peer-review our manuscript and offer valuable feedback that has greatly contributed to its improvement.
Comments:
In section 3 (Repurposing of Anthelmintic Benzimidazoles as Anticancer Agents: they mentioned benzimidazoles for other uses as follows: “Various other benzimidazole derivatives are under clinical development for other biological activities, such as anti-inflammation and anticoagulation (lines 177).” It is not appropriate for this section. It needs to be modified.
Response: We agree with the reviewer and accordingly, the paragraph has been deleted in the revised manuscript.
In section 4, to help readers understand and make To focus on the aim of this review, it needs to categorize the types of hematological cancers, describe the pathology that is problematic in each type, and then mention about the benzimidazoles that act on them and show anticancer effects.
Response: A paragraph has been added to introduce blood cancer types and cited few references related to clinical pathology based on the genetic analysis (lines 199-210). Also, we complemented the section-4 with Table-1 to summarize the studies described.
In addition, in section 4, they described the anticancer effect of benzimidazoles on other types of cancer, such as head and neck squamous cell carcinoma, glioma, melanoma, and colorectal cancer (lines 318~320). There are various reports describing the anticancer effect of benzimidazoles on another cancer type. It is not proper in this section.
Response: We thank reviewer suggestion. In the revised manuscript, studies with other cancer types is deleted. Few studies that complement the findings in blood cancer cells were discussed just to appreciate and understand the anticancer activity of benzimidazoles.
In addition, Figure 4 needs to be updated. It is recommended to include the contents, which were introduced in the section 5 (Mechanism of Action of Benzimidazoles against Hematological Cancers) and to include what kind of benzimidazole derivative acts on each mechanism and how they affect (inhibit or activate).
Response: As per reviewer suggestion, Figure-4 has been revised to indicate the name of benzimidazole along with their affect (inhibition or activation) for each of the mechanism discussed in section-5.
It would be better to summarize the contents described in section 4 of Table as recommended purpose of Anthelmintic Benzimidazoles.
Response: As per reviewer suggestion, Table-1 has been added in the revised manuscript to describe the contents discussed in Section-4.
Abbreviations should be used after defining the full name of some words, and then keep using abbreviations. However, they mix the full names and abbreviations, and sometimes they define some words again.
Response: We appreciate the reviewer’s feedback regarding the use of abbreviations. We have thoroughly reviewed the manuscript and ensured that all abbreviations are defined, with consistent use of abbreviations thereafter. Instances of mixed usage and repeated definitions have been corrected.
It needs to include references for Figure 3 (lines 121~123).
Response: As per reviewer suggestion, we added references for Figure-3 (line-137).
please check the reference lists. For example, #11. There is no source of the article.
Response: Thanks for the suggestion. We corrected ref.11 and checked all the references.
Reviewer 2 Report
Comments and Suggestions for Authors
The review discusses the potential applications of anthelmintic benzimidazole derivatives in hematological malignancies. The structure of the review is logical, starting with a short introduction, followed by a discussion of anticancer studies with benzimidazole derivatives, studies more specifically in hematological cancers and an assessment of the potential mechanistic pathways implicated. The manuscript is based on relevant references and its topic could be of interest to readers of the journal.
Comments, suggestions:
-in the introduction the choice of the topic of hematological malignancies in relation with benzimidazoles should be further substantiated
-it would be of interest to readers if the authors could comment more explicitely in the conclusion, future perspectives section on the human anticancer (clinical) data available and the likeliness of future clinical developments, main issues of this direction
-Figures 1-3 are rather generic, while no figures were used for the bulk of the discussion. It could help readers, if some of the most important mechanistic pathways implicated could be represented in more detail by some additional figures.
-It could help readers if the authors could summarize in a table format the benzimidazole derivatives discussed with their main effects + major mechanistic pathways, the main references, the types of studies (in vivo/in vitro, cell line or animal model used, cis any human data available).
Comments on the Quality of English LanguageThe quality of English language is fine, there are some minor issues/comments in this regard.
Comments, suggestions:
-Figure 2,4: I would suggest to use upper indexes in R1, R2, R3
-line 277: the „Moreover, FBZ displayed 14.5-fold selectivity in killing HL60 cells over human bone marrow stem cells at a time point of 72 h after 1 day within a short period of 3 days [31].” sentence is not fully clear – please consider rewording it to make it easier to understand what the 72 h vs the 1 and 3 days periods is referring to
-line 330: „using ABZ in combination with” – please complete with the name of the combination agent
-line 339: please check „(s (OPM2”
-line 473: please correct „Furthermore, in a recent study by Will Castro et al., demonstrated”
-line 576: „GLI transcription factors (TFs) representing the main effectors of the HH signaling pathway.” – what is the statement verb?
Author Response
The manuscript is based on relevant references and its topic could be of interest to readers of the journal.
Response: We thank reviewer for acknowledging the potential interest of the topic to the journal’s readers. We believe the manuscript offers valuable insights to the field and appreciate your positive feedback.
Comments:
In the introduction the choice of the topic of hematological malignancies in relation with benzimidazoles should be further substantiated
Response: As per reviewer suggestion, we added lines 85-90 in the introduction to further substantiate the selection of hematological cancers and benzimidazoles.
It would be of interest to readers if the authors could comment more explicitely in the conclusion, future perspectives section on the human anticancer (clinical) data available and the likeliness of future clinical developments, main issues of this direction
Response: Thank you for the suggestion. In response, we have added the relevant clinical study information in lines 85-88, 170-174, and 700-702, and have cited the appropriate references in the revised manuscript.
Figures 1-3 are rather generic, while no figures were used for the bulk of the discussion. It could help readers, if some of the most important mechanistic pathways implicated could be represented in more detail by some additional figures.
Response: As per reviewer suggestion, we revised the Figure-4 to describe the mechanistic pathways and discussed/cited in section-5.
It could help readers if the authors could summarize in a table format the benzimidazole derivatives discussed with their main effects + major mechanistic pathways, the main references, the types of studies (in vivo/in vitro, cell line or animal model used, cis any human data available).
Response: As per reviewer suggestion, Table-1 has been added in the revised manuscript to describe the contents discussed in Section-4.
Minor Comments on the Quality of English Language:
-Figure 2,4: I would suggest to use upper indexes in R1, R2, R3
Response: Corrected.
-line 277: the „Moreover, FBZ displayed 14.5-fold selectivity in killing HL60 cells over human bone marrow stem cells at a time point of 72 h after 1 day within a short period of 3 days [31].” sentence is not fully clear – please consider rewording it to make it easier to understand what the 72 h vs the 1 and 3 days periods is referring to
Response: Corrected.
line 330: „using ABZ in combination with” – please complete with the name of the combination agent
Response: Corrected.
line 339: please check „(s (OPM2”
Response: Corrected.
line 473: please correct „Furthermore, in a recent study by Will Castro et al., demonstrated”
Response: Corrected.
line 576: „GLI transcription factors (TFs) representing the main effectors of the HH signaling pathway.” – what is the statement verb?
Response: Corrected.
Round 2
Reviewer 1 Report
Comments and Suggestions for Authors
It was improved according to the suggestion.